# Subjective and Objective Quality-of-Experience Assessment for 3D Talking Heads

Yingjie Zhou
Zicheng Zhang
PengCheng Laboratory
Shenzhen, China
Shanghai Jiao Tong University
Shanghai, China
{zyj2000,zzc1998}@sjtu.edu.cn

Wei Sun
Xiaohong Liu
PengCheng Laboratory
Shenzhen, China
Shanghai Jiao Tong University
Shanghai, China
{sunguwei,xiaohongliu}@sjtu.edu.cn

Xiongkuo Min†
Guangtao Zhai†*
PengCheng Laboratory
Shenzhen, China
Shanghai Jiao Tong University
Shanghai, China
{minxiongkuo,zhaiguangtao}@sjtu.edu.cn

## ABSTRACT

In recent years, immersive communication has emerged as a compelling alternative to traditional video communication methods. One prospective avenue for immersive communication involves augmenting the user's immersive experience through the transmission of three-dimensional (3D) talking heads (THs). However, transmitting 3D THs poses significant challenges due to its complex and voluminous nature, often leading to pronounced distortion and a compromised user experience. Addressing this challenge, we introduce the 3D Talking Heads Quality Assessment (THQA-3D) dataset, comprising 1,000 sets of distorted and 50 original TH mesh sequences (MSs), to facilitate quality assessment in 3D TH transmission. A subjective experiment, characterized by a novel interactive approach, is conducted with recruited participants to assess the quality of MSs in THQA-3D dataset. Leveraging this dataset, we also propose a multimodal Quality-of-Experience (QoE) method incorporating a Large Quality Model (LQM). This method involves frontal projection of MSs and subsequent rendering into videos, with quality assessment facilitated by the LQM and a variable-length video memory filter (VVMF). Additionally, tone-lip coherence and silence detection techniques are employed to characterize audio-visual coherence in 3D MS streams. Experimental evaluation demonstrates the proposed method's superiority, achieving state-of-the-art performance on the THQA-3D dataset and competitiveness on other QoE datasets. Both the THQA-3D dataset and the QoE model have been publicly released at https://github.com/zyj-2000/THQA-3D.

## CCS CONCEPTS

• **Information systems** → **Multimedia databases**; *Multimedia streaming*; Multimedia content creation.

*Corresponding authors†.

## KEYWORDS

Quality-of-Experience, Quality Assessment, Talking Heads, Mesh, Large Quality Model, SyncNet, Multi-Modal

**ACM Reference Format:**
Yingjie Zhou, Zicheng Zhang, Wei Sun, Xiaohong Liu, and Xiongkuo Min†, Guangtao Zhai†. 2024. Subjective and Objective Quality-of-Experience Assessment for 3D Talking Heads. In *Proceedings of the 32nd ACM International Conference on Multimedia (MM '24), October 28-November 1, 2024, Melbourne, VIC, AustraliaProceedings of the 32nd ACM International Conference on Multimedia (MM'24), October 28-November 1, 2024, Melbourne, Australia.* ACM, New York, NY, USA, 10 pages. https://doi.org/10.1145/3664647.3680964

## 1 INTRODUCTION

After years of persistent efforts, significant advancements have been achieved in image and video-related domains [17, 28, 35]. Video communication, offering not only audio transmission but also real-time visual presentation, has substantially enhanced user experiences during communication, emerging as a pivotal achievement in societal interaction. Nevertheless, traditional video communication is constrained by viewpoint limitations, restricting the visual information conveyed to users. Consequently, the growing demand for immersive communication methods underscores the need for advancements in communication and media technologies. Within this context, three-dimensional (3D) talking head (TH) sequences are considered as a viable and promising technical solution to further enhance the immersion and interactivity of face-to-face communications. 3D THs offer users a wider array of viewing angles compared to traditional video communication, fostering a more immersive communication experience. However, the inherent voluminous nature of 3D THs necessitates artificial compression before transmission. Furthermore, the real-time dynamics of the transmission channel introduce delays, latency, and code stream transformations, exacerbating the challenges associated with 3D TH transmission. Both artificial processing and channel dynamics impact the communication experience of users significantly. Consequently, effective assessment and monitoring of 3D TH communication quality at user terminals hold paramount importance for optimizing communication system designs and enhancing user experiences.

However, conducting related quality assessment tasks poses significant challenges. On one hand, the production and collection of 3D THs entail substantial time and financial resources, resulting in few relevant datasets being available and impeding further development. On the other hand, the intricate data structures and large

**Table 1: The comparison of 3D quality assessment databases and proposed database.**

| Database | Type | Models | Distortions | Description |
|---|---|---|---|---|
| BASICS [1] | PCQA | 1,494 | GPCC, VPCC, GeoCNN [31] | Humans, Animals, Architectures, Landscapes |
| WPC [21] | PCQA | 740 | GPCC, VPCC, Gaussian noise, Downsampling | Fruit, Vegetables, Tools |
| SJTU-PCQA [49] | PCQA | 420 | Octree, Downsampling, Noise | Humans, Statues |
| LS-PCQA [23] | PCQA | 1,080 | Downsampling, Noise, GPCC, VPCC | Animals, Humans, Vehicles, Daily objects |
| DHH-QA [63] | MQA | 1,540 | Noise, JPEG, Downsampling, Quantization | Scanned Real Human Heads |
| DDHQA [62] | MQA | 800 | Noise, JPEG, Downsampling, Quantization, Motion Distortions | Dynamic 3D Digital Human |
| SJTU-H3D [58] | MQA | 1,120 | Noise, JPEG, Downsampling, Quantization | Static 3D Digital Humans |
| 6G-DTQA [66] | MQA (QoE) | 400 | JPEG, Downsampling, Quantization, Stall, Rebuffer | Digital Twins Transmitted Under 6G Networks |
| **THQA-3D (Proposed)** | **MQA (QoE)** | **1,000** | **Quantization, Stall, Rebuffer, Conversion, Synchronization** | **Scanned Real Human Heads** |

data volumes characterizing 3D data impose rigorous demands on existing communication systems, computational processing infrastructure, and algorithmic frameworks.

In this study, we address the challenge of data acquisition by leveraging existing resources. Specifically, we utilize 10 TH models sourced from the MultiFace dataset [48] introduced by Meta, employing subsequent processing to generate 1,000 distorted and 50 original mesh sequences (MSs). This endeavor culminates in the establishment of the 3D Talking Heads Quality Assessment (THQA-3D) dataset. Subsequently, we conduct a series of subjective and objective experiments on THQA-3D dataset for further research and analysis. The subjective evaluation entails the participation of volunteers in a well-organized manner, where the mean opinion score (MOS) serves as the definitive metric for assessing MSs' quality. On the other hand, we introduce a projection-based Quality-of-Experience (QoE) approach for the objective evaluation, simplifying the processing and quality assessment of mesh streams. Furthermore, the proposed method augments multimodal quality sensing capabilities through the incorporation of tone-lip coherence features and a large quality model (LQM). Experimental results demonstrate the superior performance of the proposed method compared to existing objective evaluation techniques on the THQA-3D dataset and representative QoE dataset. Consequently, the contributions of this paper are threefold:

- The THQA-3D dataset is established for quality assessment for the 3D THs. This dataset simulates 7 common types of distortion during 3D streaming media transmission, providing a total of 1000 distorted MSs. Besides, the 50 original MSs are also provided in the proposed dataset.
- A user-interactable subjective quality evaluation method is designed for 3D streaming media containing audio. Under the condition of using a 2D monitor, the user can shift the viewpoint to observe the 3D streaming media by using the mouse.
- A multimodal QoE method assisted by LQM is proposed. In addition, the variable-length video memory filter (VVMF) proposed in this paper can effectively improve the visual perception of 3D MSs.

## 2 RELATED WORK

### 2.1 Video Communication

In recent decades, advancements in communication technologies have significantly facilitated both industrial production and daily life activities. Particularly noteworthy is the refinement of video communication technology, which has enabled seamless real-time

interactions, ranging from individual video calls to group web conferences. However, despite these advancements, certain limitations inherent to video communication have become increasingly apparent. Primarily, the 2D nature of video media fails to capture the depth and 3D information present in face-to-face interactions, thereby limiting the fidelity of visual experiences during video communication. Additionally, the constrained viewpoint of video capture devices restricts users to passive reception of content from specific angles, precluding the provision of truly immersive communication experiences. Furthermore, the limited interactivity options offered by traditional video communication platforms further diminish user engagement and satisfaction levels. Consequently, contemporary communication expectations extend beyond real-time interaction, with users now seeking more immersive and interactive experiences.

In response to these evolving demands, immersive real-time video communication has emerged as a prominent trend in the field [12, 29, 50]. Although still in its nascent stage, immersive video communication is commonly associated with the transmission of 3D data, representing a departure from conventional 2D video streams. Within this context, two prevalent 3D data structures, namely point clouds and meshes, hold particular promise for immersive video applications. While point clouds offer a simpler data representation using discrete points to convey the position and color of 3D objects [55, 57, 65, 70], meshes provide a more detailed characterization, delineating each polygonal facet through vertices, edges, and normals [5, 64, 69]. Consequently, meshes are chosen in this study as the preferred data structure for modeling 3D THs, enabling a more accurate depiction of facial features and expressions.

### 2.2 3D Quality Assessment & QoE

The exploration of 3D data has emerged as a focal point of contemporary research [2, 39, 53, 59]. High-fidelity 3D data not only enhances subjective visual experiences for users but also facilitates computational processing in critical downstream tasks such as 3D detection and segmentation. Consequently, the evaluation of 3D data quality has become imperative. Presently, quality assessment methodologies for 3D data are categorized based on the type of data, including point cloud quality assessment (PCQA) [54, 55, 57, 60, 65] and mesh quality assessment (MQA) [61, 64, 69]. Table 1 summarizes relevant datasets in this domain, yielding several notable observations. Firstly, both point clouds and meshes are susceptible to various quality degradations during transmission and processing. Notably, noise, compression, and downsampling are prevalent distortions encountered in point clouds, whereas meshes exhibit a broader spectrum of distortion types. Besides, existing quality assessment datasets predominantly focus on single static point

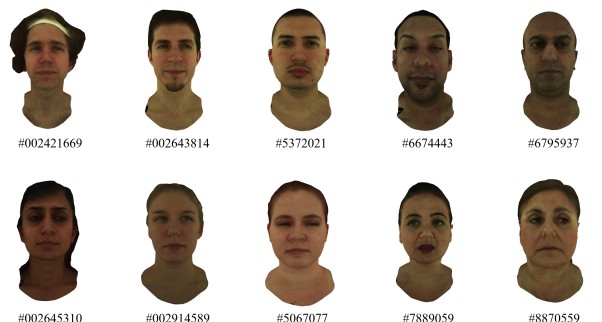

Figure 1: Overview of the selected human heads. The ID below the model remains the same as in MultiFace.

Table 2: Speech ID and the corresponding sentences.

| Speech ID | Total Sentences |
|---|---|
| DTMC | Do They Make Class biased decisions |
| GCYS | Go Change Your Shoes before you turn around |
| IDCT | If Dark Came They would lose her |
| IHTC | Its Healthier To Cook without sugar |
| TSBP | The Small Boy Put the worm on the hook |

clouds or meshes, with limited consideration given to dynamic 3D sequences.

On the other hand, while many streaming media datasets and QoE quality evaluation algorithms have been proposed, QoE quality evaluation for 3D streaming media has received little attention. Although Zhang et al. introduced the 6G-DTQA dataset [66] for dynamic digital twin QoE assessment under 6G network communication conditions, its scope remains constrained, featuring a small number of personas and a limited dataset volume. Moreover, the synthetic nature of the dynamic digital twin images, generated using artificial intelligence (AI), introduces discrepancies compared to real captured character dynamics. Notably, crucial distortion types such as code-stream transformation and audio-visual synchronization are overlooked in the 6G-DTQA dataset, despite their substantial impact on user experience. In contrast, the proposed THQA-3D dataset presented in this study leverages real-time dynamic THs captured through sensors, comprehensively simulating major distortion types affecting QoE. This includes quantization distortion, jamming, buffering, code-stream transformations, and audio-video synchronization. Through comparative analysis with existing datasets, the THQA-3D dataset emerges as a comprehensive resource for guiding QoE quality assessment in 3D data transmission scenarios.

## 3 DATABASE CONSTRUCTION

### 3.1 Source Models Collection

The THQA-3D dataset proposed in this study draws its original TH models from the MultiFace dataset [48] introduced by Meta. Thus, a succinct overview of the MultiFace dataset is provided as follows. The MultiFace dataset comprises 13 distinct identities, each represented by high-quality textured mesh heads. Utilizing a multi-camera capture studio named Mugsy, facial expressions and details are captured from multiple perspectives at a frame

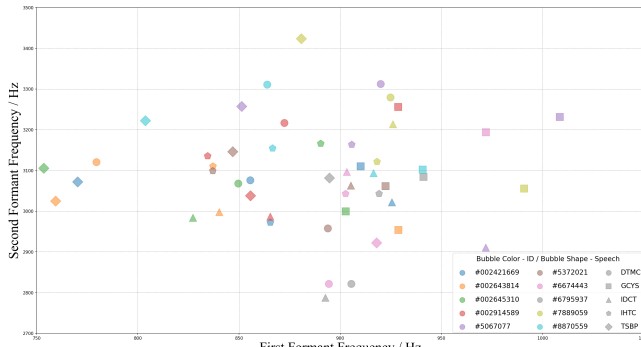

Figure 2: Phonetic attributes of the selected audio.

Table 3: Summary of the generated distortions.

| Type | Distortions | Description |
|---|---|---|
| Discontinuity | BU | Buffer at the Beginning |
| | ST | Stuck in the Middle |
| Encoding Parameters | PQ | Position Quantization |
| | TQ | Texture Coordinate Quantization |
| Code Rate | BR | Bit Rate Change |
| Synchronicity | AO | Audio Overdrive Video |
| | VO | Video Overdrive Audio |

rate of 30 frames per second (FPS). Notably, MultiFace offers high-fidelity streams of MSs and texture streams, each texture with a resolution of 1024×1024, for the 13 identities. These sequences depict individuals narrating 50 sentences of speech, accompanied by corresponding audio files. As depicted in Fig. 1, ten individuals (comprising 5 males and 5 females) are selected as raw data to construct the THQA-3D dataset. It is pertinent to mention that each talking head model is characterized by 6,172 vertices and 12,294 facets.

Furthermore, for each person, as delineated in Table 2, five fixed voices are chosen from a total of 50 voices for the QoE quality assessment study. To ensure the representativeness of the selected voices, a phonological analysis is conducted on the 50 selected voices. Specifically, the cepstrum method is employed to estimate formant frequencies for each speech, the results of which are depicted in Fig 2. The analysis reveals several key insights: Firstly, the first formant frequencies of the 50 speech sounds range from 750Hz to 1050Hz, while the second formant frequencies range from 2750Hz to 3450Hz, indicating comprehensive distribution across both formant frequencies and rich phonological features. Moreover, across different subjects, the speech of the same sentence exhibits greater similarity in formant frequencies, reflecting a certain degree of pronunciation consistency. However, variations in pronunciation consistency are observed across different sentences, indicative of individual idiosyncrasies in pronunciation habits. Overall, the 50 selected speech sounds exhibit rich phonological features, encompassing individual differences while maintaining consistency in word pronunciation, thus affirming their representativeness.

### 3.2 Distortion Generation

As indicated in Table 3 and Fig. 3, during the establishment of the THQA-3D dataset, we systematically consider potential distortions

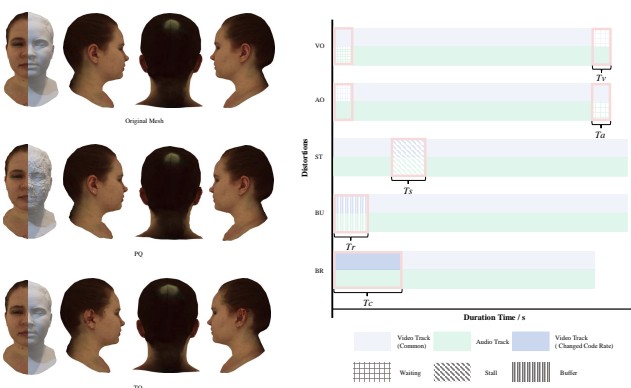

**Figure 3: An overview of the effects produced by generating distortion. On the left is a visualization of the effects of encoding parameter distortion, and the effects of streaming-related distortion are depicted on the right.**

affecting the MSs from four distinct perspectives: namely, the continuity of the MSs, the different encoding parameters, the random change of the code stream, and the audio-visual consistency.

*3.2.1 Discontinuity.* Prior research [16, 30, 36] has elucidated buffering and lagging as prominent factors impacting the quality of video streaming. Consequently, these distortions are equally pertinent in modeling discontinuities within 3D MSs. Specifically, to simulate initial buffering effects, a buffering time parameter $T_r \in [1, 2]$ is introduced, indicating the duration of buffering preceding the onset of the 3D MSs. Subsequently, the first mesh model is replicated $T_r \times FPS$ times to emulate the buffering period. Similarly, for stuck incidents, a parameter $T_s \in [1, 2]$ denotes the duration of stuck episodes. Given that the shortest speech segment lasts only 3 seconds, all stuck events are introduced after the initial 2 seconds to ensure uniform addition. Thus, mesh models at the 2-second mark are replicated $T_s \times FPS$ times and inserted following the 2-second mark to simulate stuck effects consistently.

*3.2.2 Encoding Parameters & Code Rate.* Similar to conventional communication systems, the encoding and decoding processes are indispensable for transmitting 3D MSs, inevitably subjecting 3D contents to varying degrees of distortion. Notably, the choice of encoding parameters holds significant sway over user experience, particularly in the case of mesh representations. To explore this phenomenon, this study selects two common distortions of coding parameters: position quantization and texture coordinate quantization. These distortions are simulated using the Draco[1], generating four distinct levels of coding parameter distortion. Specifically, the position quantization level is set as $q_p \in [6, 7, 8, 9]$ while the texture quantization level is set as $q_t \in [4, 5, 6, 7]$. Besides, in most adaptive code rate systems, the transmitting end adjusts coding parameters in real time based on the system's instantaneous state, resulting in fluctuations in code rates. To capture these dynamics, this study models two variations: code rate boost and code rate drop. Initially, a coding parameter configuration of $q_p = 8$ and $q_t = 6$ serves as the baseline code rate. Combining the two types of coding parameters,

[1]https://github.com/google/draco

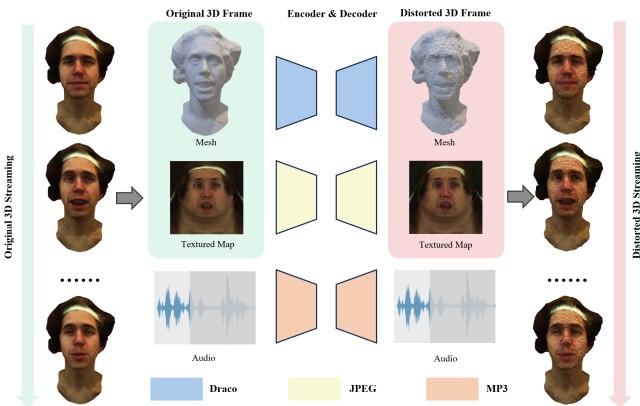

**Figure 4: Illustration of 3D Mesh Streaming Media Delivery.**

**Table 4: Details of the selected TH MSs.**

| ID | Gender | Number of Meshes | | | | |
|---|---|---|---|---|---|---|
| | | DTMC | GCYS | IDCT | IHTC | ISBP |
| #002421669 | | 93 | 115 | 124 | 113 | 99 |
| #002643814 | | 89 | 93 | 79 | 82 | 96 |
| #5372021 | Male | 149 | 165 | 123 | 141 | 139 |
| #6674443 | | 107 | 131 | 75 | 113 | 122 |
| #6795937 | | 142 | 117 | 135 | 118 | 130 |
| #002645310 | | 144 | 134 | 127 | 137 | 146 |
| #002914589 | | 104 | 102 | 100 | 98 | 96 |
| #5067077 | Female | 111 | 101 | 109 | 109 | 135 |
| #7889059 | | 116 | 123 | 118 | 116 | 101 |
| #8870559 | | 163 | 140 | 114 | 122 | 140 |

$q_p$ and $q_t$, four code rate change levels are devised and expressed by $C_r \in [-2, -1, +1, +2]$, where a positive sign indicates an increase in both types of coding parameters, while a negative sign indicates a decrease in both types of coding parameters.

*3.2.3 Synchronicity.* Due to potential blocking and delays inherent in the communication process, instances of audio-visual asynchrony may manifest in transmitted MSs. However, human perceptual sensitivity to such audio-visual misalignments is notably high. Consequently, this study endeavors to simulate two types of audio-visual misalignment effects: video overdrive and audio overdrive. These two forms of misalignment are characterized by two distinct levels, denoted by $T_v \in [0.5, 1]$ and $T_a \in [0.5, 1]$ separately.

## 3.3 3D Mesh Streaming Preparation

In the absence of a standardized protocol for 3D MSs transmission, it is imperative to delineate a communication model tailored to 3D MSs. Leveraging insights from conventional video transmission paradigms, we conceptualize each TH model as an individual 3D frame as shown in Fig. 4. Each 3D frame comprises a mesh, housing geometric information exclusively, and a texture map component, encapsulating texture and color. The sequential arrangement of these 3D frames imparts temporal continuity to the otherwise static mesh. Notably, for 3D MSs incorporating audio, such as THs, inclusion of an audio file containing voice information alongside

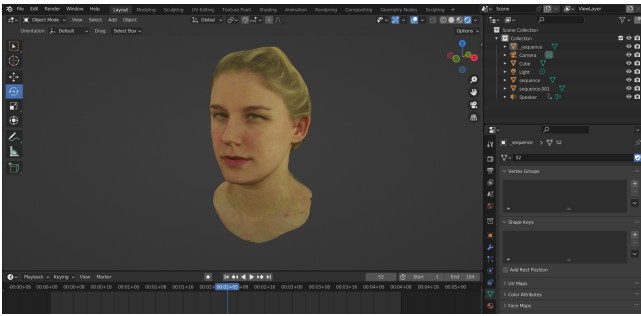

**Figure 5: Blender interface for subjective experiments. The timestamp at the bottom can be adjusted at will.**

the corresponding 3D frames is necessitated. Consequently, the mesh, texture map, and audio file are treated as distinct entities, subject to separate compression and transmission processes during communication transfer. To this end, Google's Draco is employed for mesh compression, while JPEG [24, 44] serves as the compression algorithm for texture maps, and MP3 [3] is utilized for audio compression.

On the receiving end, the transmitted stream of 3D meshes can be reconstructed through decoding of the 3D frames and audio files. The data listed in Table 4 illustrates the number of encoded 3D frames for each subject when speaking different sentences. Notably, we adhere to the original parameter configuration outlined in Multiface [48], maintaining 30 FPS during the transmission of 3D mesh streams. As a consequence, the number of 3D frames in this dataset is contingent solely upon the duration of speech segments.

### 3.4 Subjective Experiment

Given the unique characteristics of mesh streaming, a multimodal 3D streaming subjective experiment is devised, adhering to established experimental methodologies in terms of experimental settings and equipment. The subjective assessment is conducted in a meticulously controlled laboratory environment, following the guidelines outlined in ITU-R BT.500-13 [4]. An iMac monitor supporting a resolution of 4096 × 2304 pixels is utilized for the presentation of MSs, while wired headphones of superior sound quality are employed for audio perception, ensuring both a serene testing atmosphere and real-time audio.

Specifically, as illustrated in the Fig. 5, a total of 1,000 groups of distorted MSs are imported into Blender[2] using the Stop-motion-OBJ plugin[3] in advance. Additionally, the corresponding audio files are integrated into the Blender via added speakers, resulting in 1,000 Blender project files housing complete MSs. Subsequently, 26 male and 24 female participants are recruited to take part in the subjective quality assessment experiment. Different Blender projects are randomly accessed and showcased on the monitor, affording participants the freedom to manipulate viewing angles using the mouse, thereby observing the flow of MSs from diverse perspectives. Moreover, participants are afforded the flexibility to adjust timestamps for playback of the mesh stream or halt at specific 3D frames for detailed scrutiny. Throughout the playback session,

[2]https://www.blender.org/
[3]https://github.com/neverhood311/Stop-motion-OBJ

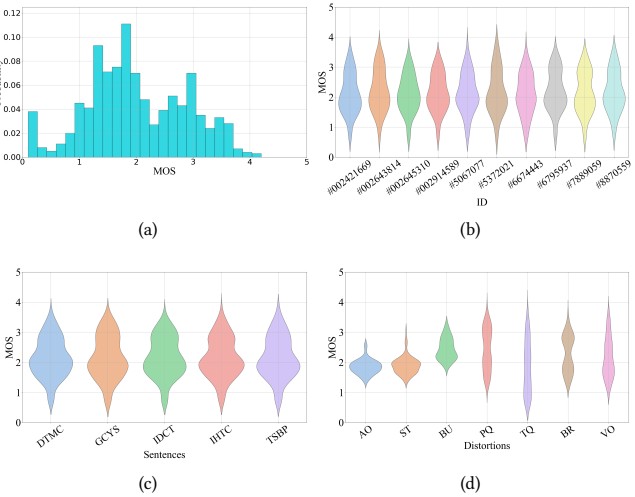

**Figure 6: Distributions of the MOSs.**

audio playback is facilitated through wired headphones. Ultimately, participants are tasked with evaluating the quality of each mesh stream in tandem with audio-visual perception. It is worthwhile to additionally note that this subjective experiment uses Absolute Category Rating (ACR) for quality assessment.

To mitigate potential discomfort [8, 37, 67] arising from prolonged exposure to 3D content, the experiment is divided into 10 phases, each comprising 100 MSs. Participants are restricted to one phase per day to prevent overexertion. Consequently, a total of 50,000 = 50 × 1,000 subjective ratings are collected upon conclusion of the experiment.

### 3.5 Subjective Data Processing

Based on previous work [56, 58, 62, 63, 66, 68], the commonly used z-scores are computed through the collection of subjective ratings. This process can be represented as:

$$z_{ij} = \frac{r_{ij} - \mu_i}{\sigma_i}, \tag{1}$$

where $r_{ij}$ denotes the quality rating provided by the $i$-th subject on the $j$-th 3D TH stream, $\mu_i = \frac{1}{N_i}\sum_{j=1}^{N_i} r_{ij}$, $\sigma_i = \sqrt{\frac{1}{N_i-1}\sum_{j=1}^{N_i}(r_{ij} - \mu_i)}$, and $N_i$ is the number of 3D TH MSs assessed by subject $i$. Furthermore, quality ratings from unreliable subjects are discarded following the subject rejection procedure recommended in [4]. Subsequently, the obtained z-scores undergo linear rescaling to the range [0, 5]. The MOS of 3D TH stream $j$ is then computed by averaging the rescaled z-scores.

### 3.6 Subjective Data Analysis

To provide a more visual representation of the subjective experiment outcomes, a histogram is generated for the processed z-scores, as depicted in Fig. 6(a). Furthermore, to delve deeper into the influence of various factors such as the 3D head models, selected speech, and different distortion types on subjective human perception, Fig. 6(b-d) are plotted, respectively. Upon examination of

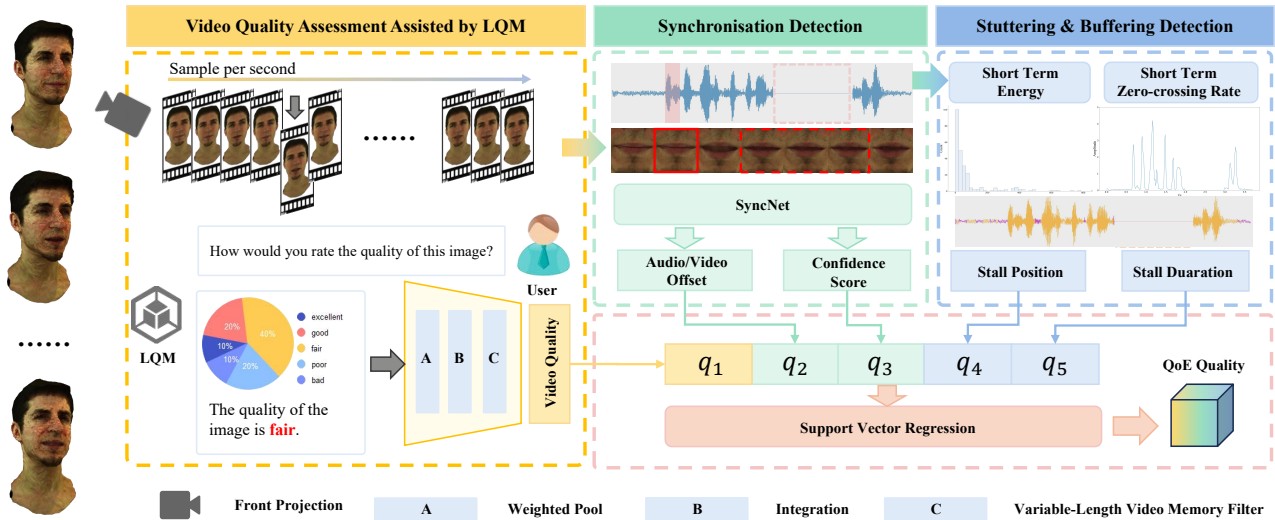

**Figure 7: The framework of the proposed QoE model.**

Fig. 6(a), it is discernible that the MOSs of the 3D TH sequences exhibit a concentration around 2.0, with fewer instances of sequences exhibiting exceptionally low or high quality. This observation implies that the distortions endured by the 3D TH sequences within the THQA-3D dataset do indeed impact human audiovisual perception to varying degrees. Conversely, the analogous distribution patterns depicted in Fig. 6 (b) and (c) suggest that subjective audiovisual perception of the 3D TH sequences exhibits no significant correlation with either the selected head models or the speech content. This finding underscores the versatility of the THQA-3D dataset, which can be effectively applied across different 3D head images and corresponding speech segments. Finally, Fig. 6(d) illustrates that distinct distortion types manifest differing quality distributions, implying that disparate forms of distortion exert distinct effects on subjective perception. This underscores the critical importance of conducting quality assessment on streaming sequences of 3D THs.

## 4 PROPOSED QOE MODEL

In this section, a novel QoE approach is proposed and details are given. The comprehensive model framework is illustrated in Fig. 7, encompassing three primary components: video quality assessment aided by the Large Quality Model (LQM), detection of audio-video consistency, and identification of discontinuities.

### 4.1 Frontal Projection

Firstly, a 3D TH composed of mesh can be represented according to prior research [69] as:

$$M \in \{\{v|v \in V\}, \{n_v|n_v \in NV\}, \{e|e \in E\}\}, \quad (2)$$

where $V$, $NV$, $E$ represent the sets of vertices, normal vectors, and edges of the mesh, respectively. Furthermore, a set of 3D TH MSs can be denoted as:

$$MS = \{M_i|i = 1, 2, ..., T_d\}, \quad (3)$$

where $T_d$ denotes the duration of the MS. As observed from the equations above, $MS$ constitutes a chronological stream of MSs

composed of $M_i$. Given the complex data structure of the mesh, direct computational operations on a stream of 3D MSs entail significant overhead. Utilizing a projection method, as demonstrated in prior studies [63, 69], offers a viable alternative. These research also indicates that the frontal aspect of a face encompasses rich facial details and expressive movements. Thus, by rendering a frontal projection of a 3D TH, it is possible to preserve the effective information contained within the 3D mesh. Consequently, the center position $C$ can be calculated for each TH:

$$C = \frac{1}{N_v} \sum_{i=1}^{N_v} v_i, \quad (4)$$

where $v_i$ represents the position of the $i$th vertex comprising the mesh, and $N_v$ denotes the total number of vertices contained within a mesh. Utilizing $C$, a 3D coordinate system can be established with the frontal face direction serving as the x-axis. Subsequently, a virtual camera is positioned at a suitable location along the x-axis to capture successive frontal projections within a mesh sequence, which are then consolidated into a single video file. Finally, audio is integrated to generate the complete video of the projected TH. The overall process can be described as:

$$V_{2D} = Fp(MS) \oplus A, \quad (5)$$

where $Fp(\cdot)$ signifies the process of front projection for each 3D frame, $A$ denotes the corresponding audio, $\oplus$ indicates the concatenation of audio and video, and $V_{2D}$ denotes the resultant 2D projected video with accompanying speech obtained through rendering.

### 4.2 Video Quality Assessment

With the increasing popularity and efficacy of large vision models (LVMs) in computer vision, these models are perceived to possess a comprehensive understanding of images surpassing that of traditional depth models. Consequently, in this study, a Large Quality Model (LQM) is employed for quality perception of video frames. Specifically, Q-Align [47] is chosen as the LQM for image quality perception. This paper selects the extracted video frames and

presents them to Q-Align with the prompt as illustrated in Fig. 7. Q-Align then returns the probability distribution across five grades: "excellent", "good", "fair", "poor", and "bad". The overall process is articulated as follows:

$$I_s^m = S(V_{2D}),$$
$$P_c^m = LQM(I_s^m), \tag{6}$$

where $S(\cdot)$ signifies the frame extraction process, $I_s^m$ represents the $m$th extracted frame sensed by the LQM to obtain the probability vector $P_c^m$ encompassing five categorical probabilities. Subsequently, a fixed weight $w_c$ is applied to the probability vector to assess the quality of the extracted video frame:

$$Q_{Is}^m = w_c \cdot P_c^m, \tag{7}$$

where $Q_{Is}{}^m$ denotes the predicted quality for the extracted frame, and $\cdot$ denotes the vector inner product. This process is iterated until all the extracted frames in the video are evaluated, resulting in a sequence of assessed frame qualities denoted as $Q_{Vs}$. To better align with human subjective perception during video viewing, a variable-length video memory filter (VVMF) incorporating an Hermann Ebbinghaus forgetting curve [10] is employed to process the quality of each frame in $Q_{Vs}$. Specifically, the simplest exponential decay is sampled discretely according to the length $L$ of $Q_{Vs}$ as:

$$w_f(n) = e^{-n}, \tag{8}$$

where $n = 0, 1, ..., L - 1$ denotes the discrete sampling time. Additionally, to constrain the filtering result, $w_f(n)$ must satisfy the following condition:

$$\sum_{n=0}^{L-1} w_f(n) = 1. \tag{9}$$

Finally, the filtering process of the VVMF is employed to derive the video quality $q_1$ using operations similar to a linear filter:

$$q_1 = \sum_{n=0}^{L-1} Q_{Is}^n w_f(L - 1 - n). \tag{10}$$

This equation can be equivalently understood as a sampling of the linear convolution of $Q_{Is}^n$ and $w_f(n)$ at the $(L - 1)$ moment, expressed as $q_1 = [Q_{Is}^n * w_f(n)]\delta(n - (L - 1))$.

### 4.3 Synchronisation Detection

The consistency between audio and video, particularly concerning mouth movements and speech, significantly influences the user's audiovisual experience. Hence, detecting synchronization between audio and video is crucial. In this study, we employ SyncNet [6], a classical feature extractor, for audio-lip consistency detection. Specifically, SyncNet conducts mouth cropping on the rendered frontal video and performs Mel Frequency Cepstral Coefficient (MFCC) feature extraction on the audio. Subsequently, consistency metrics are computed utilizing a siamese network. Within this component, the audio-video bias and confidence scores serve as features $q_2$ and $q_3$ characterizing audio-lip consistency.

### 4.4 Stuttering & Buffering Detection

During streaming, buffering or stuck events often manifest as silent intervals in the audio signal, with its amplitude remaining at zero.

This temporal pattern in the audio signal is distinctive and observable. Hence, by analyzing short-time energy and short-time over-zero rate metrics of the audio signal and setting suitable thresholds, silent intervals can be identified. Compared to traditional video-level discontinuity detection methods, this approach is more straightforward and intuitive. The onset and duration of these silent intervals are recorded as $q_4$ and $q_5$, respectively, effectively characterizing buffering or stuck during media transmission.

### 4.5 Quality Regression

Following the extraction of five features from the 3D TH frontal projection video, support vector regression (SVR) is employed for final quality assessment to streamline the algorithmic process and enhance efficiency. It is pertinent to note that the SVR model utilized in this method is based on the scikit-learn package in Python, employing a radial basis function.

## 5 EXPERIMENTS

### 5.1 Experiment Details

To validate the efficacy of the proposed Quality of Experience (QoE) model, 23 representative quality assessment methods are selected as competing algorithms. These methodologies encompass image quality assessment (IQA), video quality assessment (VQA), point cloud quality assessment (PCQA), and quality of service assessment (QoS) methods. Notably, all evaluation methods across these categories assess video quality, except for PCQA, which evaluates quality after converting meshes into point clouds. Furthermore, based on the presence or absence of reference information, the evaluation methods utilized in this experiment can be classified into full reference (FR) and no reference (NR) quality evaluation methods.

All competing algorithms are evaluated on both the proposed THQA-3D dataset and the classic Waterloo-III dataset [9]. For data partitioning, a five-fold cross-validation strategy is employed. Besides, there is no overlap in the content of all divided folds. Particularly for the THQA-3D dataset, it is important to note that Q-Align is fine-tuned with low-rank decomposition (LoRA) [14] using video frames within the coded parameter-distorted videos in each training set. However, for the Waterloo-III dataset, given the absence of audio in the Waterloo-III dataset, audio-lip consistency is replaced with video bitrate, which is a common feature in traditional QoE methods. Furthermore, experiments regarding buffering and stuck characteristics are conducted directly using the provided dataset information. Additionally, the Q-Align is fine-tuned with LoRA using all video frames in each training set. During performance testing on both datasets, the average performance over five iterations is computed to represent the algorithm's performance. Finally, it is worth stating that all methods use the source code provided by the authors and keep the original parameter settings.

### 5.2 Experiment Criteria

In order to quantify the performance of each algorithm, four widely used performance metrics are utilized, namely Spearman Rank Correlation Coefficient (SRCC), Kendall's Rank Correlation Coefficient (KRCC), Pearson Linear Correlation Coefficient (PLCC), and Root Mean Squared Error (RMSE). SRCC and KRCC primarily measure

**Table 5: Performance results on the proposed database. Best in RED, second in BLUE. NaN means not a number, which indicates that the method predicts a fixed constant mass result, indicating that the method is invalid.**

| Type | Models | Ref | THQA-3D | | | | Waterloo-III | | | |
|---|---|---|---|---|---|---|---|---|---|---|
| | | | SRCC↑ | PLCC↑ | KRCC↑ | RMSE↓ | SRCC↑ | PLCC↑ | KRCC↑ | RMSE↓ |
| IQA | PSNR | FR | 0.3485 | 0.4203 | 0.2493 | 0.7838 | 0.2001 | 0.4502 | 0.1271 | 14.6370 |
| | SSIM [45] | FR | 0.5438 | 0.6012 | 0.4140 | 0.6862 | 0.2413 | 0.4467 | 0.1556 | 14.6662 |
| | NIQE [26] | NR | 0.2243 | 0.4741 | 0.1232 | 0.7707 | 0.2983 | 0.5274 | 0.2070 | 12.9010 |
| | IL-NIQE [52] | NR | 0.2293 | 0.4871 | 0.1537 | 0.7600 | 0.1931 | 0.4097 | 0.1298 | 13.8712 |
| VQA | VMAF [20] | FR | 0.2598 | 0.2907 | 0.1920 | 0.8322 | 0.4672 | 0.6231 | 0.3338 | 11.0234 |
| | VIIDEO [25] | NR | 0.1056 | 0.2308 | 0.0721 | 0.8387 | 0.1448 | 0.1164 | 0.1189 | 14.9789 |
| | TLVQM [15] | NR | 0.1887 | 0.3112 | 0.1272 | 0.8240 | 0.0452 | 0.1352 | 0.0321 | 15.2116 |
| | VIDEVAL [42] | NR | 0.2252 | 0.3544 | 0.1556 | 0.8118 | 0.1822 | 0.1284 | 0.1317 | 16.2566 |
| | RAPIQUE [43] | NR | 0.3748 | 0.4680 | 0.2660 | 0.7643 | 0.0390 | 0.2828 | 0.0305 | 15.1513 |
| | SimpVQA [38] | NR | 0.6321 | 0.7258 | 0.4717 | 0.5983 | 0.6083 | 0.6207 | 0.4096 | 10.4651 |
| | VSFA [19] | NR | 0.7463 | 0.7811 | 0.5596 | 0.5726 | 0.4177 | 0.3669 | 0.3664 | 11.8469 |
| | FAST-VQA [46] | NR | 0.7778 | 0.7984 | 0.5964 | 0.5503 | 0.7515 | 0.7332 | 0.5469 | 10.0576 |
| | BVQA [18] | NR | 0.7871 | 0.8298 | 0.6081 | 0.5983 | 0.7682 | 0.7126 | 0.5602 | 10.6440 |
| PCQA | PSNRyuv [41] | FR | 0.2810 | 0.0739 | 0.2063 | 0.8673 | - | - | - | - |
| | MSE-p2po [7] | FR | 0.0374 | 0.3060 | 0.0256 | 0.8284 | - | - | - | - |
| | MSE-p2pl [40] | FR | 0.0371 | 0.2710 | 0.0257 | 0.8703 | - | - | - | - |
| QoS | P.1203 [32, 33] | NR | 0.0808 | 0.1186 | 0.0502 | 0.8597 | 0.2504 | 0.2966 | 0.1741 | 14.5293 |
| | FTW [13] | NR | 0.0515 | 0.0658 | 0.0356 | 0.8586 | 0.2238 | 0.4106 | 0.1572 | 13.9931 |
| | Liu2012 [22] | NR | 0.0562 | 0.0425 | 0.0465 | 0.8606 | 0.5082 | 0.6461 | 0.3645 | 11.5858 |
| | Mok2011 [27] | NR | 0.0518 | 0.0853 | 0.0416 | 0.8762 | 0.2399 | 0.3252 | 0.1887 | 14.2519 |
| | Yin2015 [51] | NR | 0.0416 | 0.0624 | 0.0255 | 0.8582 | 0.4502 | 0.4104 | 0.3250 | 13.8739 |
| | VsQM [34] | NR | 0.0490 | 0.0613 | 0.0412 | 0.8583 | 0.2504 | 0.2966 | 0.1741 | 14.5293 |
| | PXNR [11] | NR | 0.2482 | 0.4352 | 0.1671 | 0.7823 | NaN | NaN | NaN | 15.3660 |
| | **Proposed** | NR | 0.8411 | 0.8434 | 0.6567 | 0.4501 | 0.7888 | 0.8147 | 0.6053 | 10.2825 |

the monotonicity of prediction quality, while PLCC assesses the linearity and consistency of the algorithm's prediction quality. RMSE is employed to evaluate the accuracy of the prediction results. When SRCC, PLCC and KRCC approach 1, and RMSE approaches 0, it indicates excellent performance of the algorithm.

## 5.3 Performance Analysis

The experimental results of the performance tests are presented in Table 5. By analyzing Table 5, the following conclusions can be drawn: 1) The proposed algorithm achieves state-of-the-art performance on the THQA-3D dataset, significantly outperforming the other algorithms involved in the experiments. 2) The performance of the proposed method on the Waterloo-III dataset remains competitive. However, considering the specific experimental setup on the Waterloo-III dataset, the results simply demonstrate that the large model-assisted QoE approach remains effective for classical QoE datasets. 3) Comparison of the experimental results on the THQA-3D and Waterloo-III datasets shows that some of the methods involved in the comparison are not stable, and even PXNR shows a complete failure result. This also highlights that the proposed method has stability. 4) The representative methods selected in the experiments exhibit inadequate performance in handling the challenge of streaming 3D TH MSs. This is mainly attributed to the insensitivity of traditional IQA, VQA, PCQA, and QoS methods to distortions caused by audio modalities, despite their comprehensive assessment of stream quality from various perspectives.

## 5.4 Ablation Experiments

To further explore the individual contributions of each module, ablation experiments are conducted on the THQA-3D dataset. The results of these experiments are presented in Table 6. By analyzing Table 6, the following conclusions can be drawn: 1) Large model-assisted video quality sensing, audio-lip consistency detection, speech-based buffering and stuck detection, and VVMF all

**Table 6: Ablation study results on THQA-3D database, where 'w/o' stands for 'without'. Best in RED.**

| Model | VVMF | SRCC↑ | PLCC↑ | KRCC↑ | RMSE↓ |
|---|---|---|---|---|---|
| w/o $q_1$ | - | 0.4176 | 0.5212 | 0.2967 | 0.7225 |
| w/o $q_{2\sim3}$ | ✗ | 0.6212 | 0.6984 | 0.4688 | 0.6117 |
| | ✓ | 0.6390 | 0.7114 | 0.4833 | 0.6003 |
| w/o $q_{4\sim5}$ | ✗ | 0.7320 | 0.7519 | 0.5462 | 0.5654 |
| | ✓ | 0.7578 | 0.7723 | 0.5701 | 0.5457 |
| $q_1$ | ✗ | 0.4761 | 0.6450 | 0.3358 | 0.6562 |
| | ✓ | 0.5198 | 0.6600 | 0.3730 | 0.6461 |
| $q_{2\sim3}$ | - | 0.3659 | 0.5019 | 0.2575 | 0.7388 |
| $q_{4\sim5}$ | - | 0.0974 | 0.2489 | 0.0742 | 0.8344 |
| All | ✗ | 0.8225 | 0.8299 | 0.6351 | 0.4664 |
| | ✓ | 0.8411 | 0.8434 | 0.6567 | 0.4501 |

positively contribute to the overall algorithm performance, indicating the effectiveness of all modules. 2) From an importance standpoint, video perception is the most crucial, followed by lip-sound consistency, while buffering and stuck detection play a relatively minor role. This is due to the intuitive nature of visual information acquisition and its dominance in human sensory perception. Additionally, humans are highly sensitive to audio-visual asynchrony, while buffering and lagging phenomena indirectly affect viewer emotions. Thus, the proposed algorithm aligns with the principles of human subjective perception. 3) Analyzing the image quality level, the performance of the large model-assisted quality perception module surpasses that of existing NR IQA methods, indicating the large model's ability to perceive more in-depth image quality features and laying a solid foundation for overall video quality perception.

## 6 CONCLUSIONS

This paper contributes to the field of immersive video communication by introducing the 3D Talking Heads Quality Assessment (THQA-3D) Dataset. This dataset encapsulates seven types of distortions commonly encountered in 3D streaming communication, comprising 1,000 distorted 3D mesh sequence (MS) streams alongside 50 original counterparts. Furthermore, a tailored subjective experimental methodology is proposed for the quality assessment of 3D MSs. This approach involves importing received 3D MSs and speech audio into Blender, enabling subjects to assess quality from various viewpoints and evaluate the overall audiovisual experience subjectively. Moreover, we present a large model-assisted multimodal fusion method for objective quality assessment. The method leverages Q-Align, a cutting-edge large quality model, for image perception, and incorporates a variable-length video memory filter designed to integrate forgetting curves to derive video quality scores. Additionally, features extracted from speech signals are employed for audio-lip coherence analysis, and buffering and stuck detection in 3D streaming. These features are then fed into support vector regression to obtain the final predicted quality of the 3D streaming media. Experimental results demonstrate the superior performance of the proposed method. This work is anticipated to make significant contributions to the quality monitoring and evaluation of 3D streaming media, offering valuable insights for future research endeavors in this domain.

# ACKNOWLEDGMENTS

This work is supported in part by the Major Key Project of PCL (PCL2023A10-2), the China Postdoctoral Science Foundation under Grants (2023TQ0212, 2023M742298), the Postdoctoral Fellowship Program of CPSF under Grant (GZC20231618), the National Natural Science Foundation of China (623B2073, 62301316, 62101326, 62225112).

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
