# OpenReview forum: "Subjective and Objective Quality-of-Experience Assessment for 3D Talking Heads"
_acmmm.org/ACMMM/2024/Conference — MM2024 Poster_

### Official Review · Reviewer_oKYr · 2024-05-02

**Rating:** 5
**Confidence:** 2

**Summary:**

This paper proposes a new QoE Model and tests it on a new Dataset of 3D Talking Heads, which is also introduced in this paper, as well as on the Watterloo-III Dataset.
The authors compare their model with other SoA metrics on their own dataset and the Waterloo-III.
The authors show every step with all the equations of how the model is constructed and what is influencing the outcome of the model.

**Strengths:**

The strength of the paper is, that the model is described in detail and compare it to other models/metrics on two datasets.
The single steps that are shown are all easy to follow and the whole process seems well thought out.
The content in the Tables and Figures is well suited to guide the reader through the paper.

The first part of the paper about the dataset is well written and shows nicely how the dataset was created.
The second part describes the Model Creation in detail so that everyone can follow the process.
The metrics that were used to compare the model with others are a good starting point for analysis of the dataset and the model.

**Limitations:**

Besides the well planned structure of the paper it is still somewhat hard to read the paper because of all the equations and the following descriptions of the equations. This is because the description of the equations follows the same scheme over and over again (xyz denoted...)
In some cases the equation could be left out, because the text is already describing what is shown in the equation (e.g. 2 or 12).
Furthermore some descriptions of the equations are not giving additional information (e.g. in 14: SVR() denoted the SVR process (no information what the process is, and the reader would need to look that up anyways))
With leaving out some text parts that describe equations that are self explanatory, some figures like Figure2 or Figure6 could be enlarged because they are barely readable in a printed version. Maybe it is enough to enlarge the fontsize of the figures.
It would be nice if the authers would compare their dataset to other talking head datasets with objective metrics. Comparing it to the Waterloo-III is not giving the best insights to compare a talking head dataset to.

Other points:
- Typo in Table2 (heathier -> healthier)

**Suitability:**

2

---

### Official Review · Reviewer_Ca13 · 2024-05-16

**Rating:** 4
**Confidence:** 3

**Summary:**

# summary

The paper presents a dataset, subjective and objective evaluation of quality for 3D talking heads.
The model looks interesting, but there are some unclear parts in the evaluation (see below).
Furthermore, the subjective study may be a bit hard for one participants to rate 1000 3D heads, in video tests this is usually restricted to max 1 hour of testing.


## comments

### intro
* "artificial compression before transmission", why artificial, why not just compression?
* what is a "large quality model"
### database construction
* why MP3 for audio compression and not opus or AAC, why JPEG for the textures and not something more modern?

* so each participants rated 1000 3DMS?, were they paid? (and moreover any kind of demographic overview of the participants, e.g. age?)
* eq. 2, an average calculation must not nessessarily being expressed by a separate equation
* Fig 6 is not readable

### proposed qoe model
* the only question I have here, is why a SVR and not something else was used? e.g. a RF model, or a neural network?

### experiments
* "four widely used quality metrics", I would rather call them performance metrics
* "The proposed algorithm achieves state-of-the-art performance on the THQA-3D dataset", well but the proposed method is also trained for this dataset, the other models are not?, furthermore was this now the performance values of the 5 fold evaluation?, a bit unclear here.
* Table6 --> Table 6

### conclusion

**Strengths:**

* new dataset
* new model
* subjective evaluation

**Limitations:**

* evaluation and some design choices are a bit unclear

**Suitability:**

3

---

### Official Review · Reviewer_i8M8 · 2024-05-24

**Rating:** 4
**Confidence:** 2

**Summary:**

The paper presents a large-scale 3D Talking Heads Quality Assessment (THQA-3D) dataset. The dataset includes different types of distortions encountered in 3D TH transmission. Furthermore, a multimodal QoE model for quality assessment of 3D THs is proposed. The proposed model has been extensively evaluated and is shown to perform better than state-of-the-art models using a five-fold cross-validation strategy.

**Strengths:**

Strengths
1) A large-scale dataset of 3D THs with MOS from 50 subjects has been created.
2) A multimodal QoE model that is shown to outperform state-of-the-art models is proposed.
3) Detailed evaluation of the proposed model.

**Limitations:**

Weaknesses
1) The test setup is not explained in detail, especially the audio part.
2) No analysis of the viewing behaviour of the subjects and its impact on the overall MOS has been conducted. The subjects were allowed to pause and also to adjust timestamps for playback which may have an impact on the perceived quality.

**Suitability:**

3

---

### Official Review · Reviewer_9KJw · 2024-05-28

**Rating:** 2
**Confidence:** 3

**Summary:**

This paper constructs a database, conducts a subjective experiment, and proposes a no-reference objective quality metric for "3D talking heads". Although the paper tries to cover most aspects of the subjective quality assessment domain, it does so in a superficial way.  At a higher level, although the topic is somehow motivated, what are realistic use cases of "talking heads"? Why is there a need for a metric specialized on "talking heads" and on the selected (simplistic) types of distortions?

**Strengths:**

+ timely and interesting topic

The paper is timely and there are some interesting aspects, most importantly, the simultaneous evaluation of 3D models and audio in an interactive fashion. However, there is need for more in-depth description and analysis of the experimental factors, as well as their impact in the quality of experience of the users (see below).

**Limitations:**

- novelties
- reproducibility
- references
- writing

General:
- Most novelties of the paper are incremental. Specifically, the database construction relies on the selection of contents from an already existing repository, while the proposed objective quality metric is a composition of learning-based modules that are already available in the literature.

Related work:
- In section 2.2, the authors have completely neglected to refer to previous work on quality of assessment for both meshes and point clouds - both subjective and objective quality methods. Refer to [1] (i.e., it is the latest review I am aware of) and choose some of the methods discussed, which can be further enriched with newer submissions.

Subjective experiment:
- The subjective experiment is treated superficially. There are no sufficient descriptions of the environment, procedure, and methodology used, raising questions regarding its validity and reproducibility. For example, what were the testing setup and environmental conditions? What was the protocol used (e.g., ACR, DSIS)? What were the instructions regarding interactivity, and rating of audio and visual information? How was the training performed?
- There is no analysis beyond histograms for subjective scores. For example, how long and how differently did the users interact with each content? Were there any differences between different types of distortions?
- Using 3D video + audio + interactivity in the same experiment, I believe there are too many influencing factors for the rated quality from users, which makes it hard to identify where the quality variations are coming from. How do the authors comment on that, or what precautions did they take?
- In section 3.4, it is written:
"... while wired headphones of superior sound quality are employed for audio perception"
What was the exact model of the headphones used?

Objective metric:
- How realistic is the usage of the frontal view only of the mesh model in the metric? How much is the performance of the LQM affected by the inclusion of additional views?
- There is no information regarding the placement of the virtual camera on the x-axis. Was it the same for all models? Is the projected region across models the same? How much does the performance of the LQM is affected by such choices?
- It seems that each feature is dedicated to a specific type of distortion introduced in the constructed dataset. It feels as if the authors are tailoring the problem and solution to fit each other.
- It is not clear why the authors replaced proposed features (i.e., audio-lip consistency) with a different feature (i.e., video bitrate) when evaluating their metric in Waterloo-III dataset. The authors should either include the latter feature in their proposed metric or use a subset of the proposed ones.

Experiments:
- Are the selected metrics suitable for evaluating the constructed dataset? Does any of those consider audio information?
- What is the reason for using point cloud objective quality metrics on meshes? Why not use mesh metrics? How did the authors make the conversion from meshes to point clouds? Why did they not use state-of-the-art point cloud metric solutions?

Other comments:
- In section 2.2, second paragraph, please fix the first 2 sentences.
- In Table 3, with "Code Rate" I understand you mean "Bitrate". Moreover, "synchronicity" should have the first letter as capital to follow the same naming conventions. Please clarify/fix.
- In Figure 3, there are inconsistencies of terms: "BY" -> "BU", "T_v" -> "T_u"
- In section 3.1, there is a typo in the title: "Collectcion" -> "Collection"
- In section 3.3, the title is a bit misleading. The used pipeline is at best a simulation of streaming. In fact, authors just encode and decode the content.
- In section 4.2, provide a formal definition for Q_V_s.

Last but not least, across the entire manuscript, there is a selection of words that is quite unlikely to have been made by humans. This is not a plus for this paper. I would like to remind the authors that, according to ACM rules, if an LLM has been used to write the paper, it is mandatory to state so in the acknowledgments.

[1] https://doi.org/10.1016/B978-0-32-391755-1.00024-9

**Suitability:**

2

---

### Meta-Review · Area_Chair_Svrn · 2024-07-11

**Recommendation:** Accept (Poster)
**Confidence:** 4

**Metareview:**

There was a general consensus that this paper is of interest to the community. The decision is accept  but some key concerns remain and a shepherd should accompany this final acceptance.

- analysis of the viewing behaviour of the subjects and its impact on the overall should be included with some discussion.
- writing (some of it seems to be generated) - this is a MAJOR concern of the paper in its current form.
- more details on the test set up to be added.
- writing in terms of flow and figures to be improved.